# The Antimicrobial Potential and Aquaculture Wastewater Treatment Ability of Penaeidins 3a Transgenic Duckweed

**DOI:** 10.3390/plants12081715

**Published:** 2023-04-20

**Authors:** Lin Yang, Ximeng Luo, Jinge Sun, Xu Ma, Qiuting Ren, Yaya Wang, Wenqiao Wang, Yuman He, Qingqing Li, Bing Han, Yiqi Yu, Jinsheng Sun

**Affiliations:** 1Tianjin Key Laboratory of Animal and Plant Resistance, College of Life Sciences, Tianjin Normal University, Tianjin 300387, China; 2Tianjin Nankai Xiangyu School, Tianjin 300387, China

**Keywords:** duckweeds, penaeidins, bacteria inhibition, transcriptomics, quantitative proteomics

## Abstract

With the development of aquaculture, wastewater treatment and diseases have been paid more and more attention. The question of how to improve the immunity of aquatic species, as well as treat aquaculture wastewater, has become an urgent problem. In this study, duckweed with a high protein content (37.4%) (*Lemna turionifera* 5511) has been employed as a feedstock for aquatic wastewater treatment and the production of antimicrobial peptides. Penaeidins 3a (Pen3a), from *Litopenaeus vannamei*, were expressed under the control of CaMV-35S promoter in duckweed. Bacteriostatic testing using the Pen3a duckweed extract showed its antibacterial activity against *Escherichia coli* and *Staphylococcus aureus*. Transcriptome analysis of wild type (WT) duckweed and Pen3a duckweed showed different results, and the protein metabolic process was the most up-regulated by differential expression genes (DEGs). In Pen3a transgenic duckweed, the expression of sphingolipid metabolism and phagocytosis process-related genes have been significantly up-regulated. Quantitative proteomics suggested a remarkable difference in protein enrichment in the metabolic pathway. Pen3a duckweed decreased the bacterial number, and effectively inhibited the growth of *Nitrospirae.* Additionally, Pen3a duckweed displayed better growth in the lake. The study showed the nutritional and antibacterial value of duckweed as an animal feed ingredient.

## 1. Introduction

Aquaculture provides high-quality protein and delicious food for humans [1]. Chinese aquaculture production, with an annual increase rate of 7.5%, accounts for 46% of the aquacultural production in the world [2]. Aquaculture in China accounts for more than 60% of the total amount of global aquaculture. With an excessive input of fodder, the increasing nutrients pose a risk of aquatic pollution to the environment [3]. Therefore, aquacultural wastewater treatment, or making full use of wastewater, is essential. Wastewater produced by aquaculture contains high concentrations of nitrogen (N), phosphorus (P), and organic matters, including an N concentration of 2.6 mg/L and phosphorus (P) of 0.27 mg/L [1]. N is the biological substance for chlorophylls, peptides and proteins [4]. P is also crucial for lipids, proteins, and nucleic acids [5]. Thus, the aquacultural wastewater can be reused to culture plants, which is also a solution for N and P pollution.

Duckweed (*Lemnacecae*) could be applied as a feed resource, bioreactor, and an alternative stock for nutrient removal from aquacultural wastewater. It has several advantages. Firstly, the biomass of duckweed can double in 2 or 3 days compared with other plants [6], and duckweed has always been used as aquatic remediation, as it can remove N (particularly ammonia) efficiently [7]. Secondly, duckweed is also gathering interest as a raw feed material for fish, poultry, and pigs [8,9]. Duckweed promotes fish growth; for example, a duckweed-based carp polyculture system has confirmed that duckweed has positive effects on the growth rates of *Cyprinus carpio* and *Catla catla* [10]. Thirdly, duckweed is a widely distributed aquatic floating plant with high adaptability [11], leading to salt stress tolerance in transgenic duckweed *(Lemna minor*). In some warm climates, duckweed grows year-round in some water systems. Moreover, the protein content of duckweed is very high, which is suitable for expressing foreign proteins effectively [12]. Duckweed, because of its high nutritional value, could be employed as human food and animal feed [13]. Therefore, duckweed could be an ideal feedstock for wastewater removal, feed resource, and bioreactor.

Antimicrobial peptides (AMPs) play a role in resisting bacteria, viruses and other microorganisms [14], including resistance to diseases that infect aquatic organisms. The worldwide penaeid shrimp aquaculture, with a revenue value of about US $32 billion per year, has been consistently affected by devastating diseases, causing a severe loss in production [15]. Antibiotics have been suggested for controlling vibriosis in aquaculture, but with the problem of antibiotic resistance this presents risks [16,17,18]. Penaeidins, as a natural immune substance extracted from *Litopenaeus vannamei* [19], have an inhibitory effect on Gram-negative bacteria and Gram-positive bacteria due to the sequence divergence in the N-terminal proline-rich domain (PRD) and subsequent conformational differences [20,21,22]. Penaeidins within shrimp hemocytes were challenging to extract from biological samples [23].

Duckweed is an excellent candidate for penaeidins expression in the aquatic system [24]. First of all, duckweeds are practical and economical bioreactors of various types [25], especially those duckweeds breeding by vegetative propagation under long daytime conditions. The daughter fronds are produced by budding within a single pouch of the mother frond, leading to the stable genetic bioreactor. Secondly, the duckweeds can be secreted into the liquid medium, so that a more straightforward product purification will suffice [26]. Thirdly, duckweeds can be used as feed. Additionally, the expression of penaeidins improves the nutritional value of the feed. Furthermore, compared with prokaryotes such as *Escherichia coli,* duckweed, as a eukaryote, has the advantage of expressing penaeidins with higher activity. Using duckweed as a bioreactor to produce penaeidins will be an excellent solution to the major public health challenge of microbial antibiotic resistance.

Duckweed provides an advantage in expressing foreign proteins. For example, the extracellular domain of matrix protein 2 (M2e peptide) of avian influenza virus H5N1 has a high-yield expression in transgenic duckweed, promising the development of a duckweed-based expression system to produce an edible vaccine against avian influenza [25]. Additionally, duckweed is considered a promising source of protein for food products, due to its high protein content and environmentally friendly production [13]. We have explored a workable way to express the penaeidins in duckweed. The Pen3a duckweed can be both cultured in industrial conditions and the pond, and make use of the aquaculture-culturing wastewater.

Here, we report a successful penaeidins 3a transformation of duckweed. The aquaculture wastewater treatment, bacteriostatic test on the extract of penaeidins 3a transgenic duckweed, and the direct bacteriostatic activity in the liquid medium have been studied in the research. We also analyzed the transcriptome, proteomics, and microbial communities in penaeidins 3a duckweed.

## 2. Results

### 2.1. Identification of Transgenic Duckweed

Pen3a transgenic duckweeds (designated as Pen3a) were obtained by *Agrobacterium* mediated transformation, and five separate transgenic lines were cultured in a liquid medium. The plant expression vector pCAMBIA-1301-Pen3a was constructed (Figure 1a), and the GUS gene on the plant expression vector p1301 is used as a reporter gene; its activity was detected in roots and fronds of Pen3a duckweed (Figure 1b). These 5 independent transgenic lines were detected by PCR identification (Figure 1c). RNA sequence results showed that the mRNA transcripts were expressed in 5 transgenic lines, but not in the wild type (WT) duckweed (Appendix A). The readcount of *Litopenaeus vannamei* Penaeidin in the wild-type duckweed was 0 and the expression of Pen3a_K has the highest readcount among the five transgenic lines. These results indicated that penaeidin 3a had been successfully expressed in transgenic duckweed.

### 2.2. Production Analysis and Nutrient Removal Rate in Duckweed

To study the production of WT duckweed and Pen3a duckweed during aquacultural wastewater treatment, the protein production, ash, N, P, and K content in duckweed were investigated. As shown in Table 1, the protein content in Pen3a duckweed was 37.38%, 1.11% higher than the WT duckweed. The nitrogen (N), phosphorus (P) and potassium (K) were 5.98%, 1.10%, and 6.86%, respectively. The high protein production potential of Pen3a and WT duckweed can be applied as the feed protein source. The N, P, and K uptake within 7 days was calculated. Additionally, the N removal rate, P removal rate, and K removal rate were 191.15, 45.795, and 255.086 mg/m^2^/day in WT duckweed. The Pen3a duckweeds have a similar recovery rate as WT, with an N removal rate, P removal rate, and K removal rate of 191.15, 45.795, and 255.086 mg/m^2^/day, respectively. This result demonstrated a high nutrient removal capacity in both WT and Pen3a duckweed.

### 2.3. Antibactericidal Tests of Pen3a Duckweed Extracting Solution

A bacteriostatic test on the Pen3a duckweed extract and WT duckweed extract was performed. We chose *Staphylococcus aureus* and *Escherichia coli* as representatives of Gram-positive and Gram-negative bacteria, respectively. Additionally, these two bacteria could be applied to distinguish the effect of the extract on different bacteria. The antibacterial activity of Pen3a duckweed extract against the Gram-negative bacteria *Escherichia coli* and the Gram-positive *Staphylococcus aureus* was obvious. The inhibition zone of the different concentrations of extract on *E. coli* and *S. aureus* are shown in Figure 2 (the concentrations are 50%, 75% and 100%). The inhibition zone of *E. coli* was about 19.2 ± 0.6 mm when the extraction of concentration was 100% (Figure 2a) and the inhibition zone of around 15.5 ± 0.5 mm was observed for *S. aureus* (Figure 2b). For the antibacterial test, the WT duckweed extracting solution as a control has been added in the Appendix A. The WT duckweed extract showed no antibacterial activity of *E. coli* (Appendix A) and *S. aureus* (Appendix A). These results showed that Pen3a duckweed had an inhibitory effect on bacteria.

### 2.4. Effects of Aquacultural Water on WT Duckweed and Pen3a Duckweed

Plants possess innate immune systems capable of responding to bacteria and fungi. In this study, we investigated the growth of WT and Pen3a duckweed cultured in aquaculture wastewater for 14 days. Additionally, the WT duckweed and Pen3a duckweed showed different phenotypes. The roots of wild duckweed were longer than Pen3a duckweed, and the fronds of Pen3a duckweed still kept three frond groups; however, the WT frond groups were broken up (Figure 3a). According to the statistics, after 14 days of cultivation in the aquacultural wastewater, the root abscission rate of WT duckweed was 31.37%, and that of Pen3a duckweed was 20.16% (Figure 3b). The abscised roots of Pen3a duckweed were shorter than WT duckweed. Interestingly, compared with WT duckweed, the water treated by Pen3a duckweed was very clean. We measured the absorbance at 600 nm of the aquaculture wastewater after Pen3a duckweed and WT duckweed were cultured for 14 days. The absorbance at 600 nm of wastewater treated by Pen3a and WT duckweed were 0.098 and 0.265, respectively. This result showed the wastewater treatment ability of Pen3a duckweed (Figure 3c). These phenomena indicated some self-protection mechanisms of transgenic duckweed in response to adverse environments. Therefore, duckweed can be co-cultured with the wastewater from shrimp cultivation in an outdoor water recycling system, be applied to obtain penaeidins extraction and feed production, and play a role in bacteria inhibition.

### 2.5. Microbiomes with Statistical Difference

Linear discriminant analysis Effect Size (LEfSe) was used to analyze the differences in the abundance of bacteria in aquacultural water. LEfSe was suitable for the statistical analysis of metagenomic data on multiple microbiomes. From Figure 4a, we can infer the evolutionary relationship among the microorganisms and the significant differences among groups. Among the three treatments, LEfSe detected 88 bacterial clades, among which LDA values were higher than 2.5 (Figure 4b). These microorganisms mainly belonged to *Nitrospirae*, *Sphingobacteriia*, *Synechococcophycideae*, and *Opitutae*. In the control group, the genus *Nitrospirae,* which belonged to the phylum *Nitrospirae,* was substantially higher. Nevertheless, both WT and Pen3a duckweed significantly decreased the bacterial abundance and effectively inhibited the growth of *Nitrospirae*. However, compared with Pen3a duckweed, WT duckweed was cultivated in aquaculture tail water, the water body was more turbid, and the total number of bacteria was more, while Pen3a significantly reduced the number and abundance of bacteria. Duckweed has been reported to host a similar bacterial assemblage with the terrestrial leaf microbiome in taxonomy.

### 2.6. Post-Duckweed Treatment Analysis on the Bacterial Community in the Aquacultural Wastewater

The taxonomic analysis indicated that the dominant four phyla were *Proteobacteria*, *Cyanobacteria*, *Actinobacteria*, and *Bacteroidetes* after 2 different treatments (Figure 5a). Compared with the control group, the relative abundance of *Chlorobi* increased in the WT and Pen3a duckweed treatment groups. Pen3a duckweed can inhibit the growth of *Firmicutes*. After 14 days in the lake water, the abundance of *Firmicutes* after Pen3a duckweed treatment was 0.382%, while after WT duckweed treatment, it was 7.104%. To explore the similarity and differences among different groups, we constructed a cluster tree of samples to investigate the similarities and differences (Figure 5b). The results show that WT duckweed and Pen3a duckweed completely inhibited the growth of *Nitrospirae*.

### 2.7. Transcriptome Analysis of WT and Pen3a Duckweed

The gene expression of five transgenic and wild-type duckweed samples was analyzed. Overview of the expression profile of the DEGs is shown in Figure 6a. The cluster analysis between Pen3a duckweed and WT duckweed was significantly different. To understand the potential functions of DEGs in the Pen3a duckweed, gene ontology (GO) enrichment analysis was conducted (Figure 6b). Three different classifications represented three basic classifications of GO terms, biological processes (BP), cellular components (CC), and molecular functions (MF). “Biosynthetic process”, in the BP category, “cell and cell part” in the CC, and “structural molecule activity” in the MF category were the most down-regulated DEGs. Additionally, the most up-regulated DEGs in the BP section were “protein metabolic process”, and in the MF category it was “ion binding”.

### 2.8. Molecular Mechanism Changes in Growth-Related KEGG Pathway

Sphinganine acts as the signaling molecule during biotic and abiotic stresses. The expression of key enzymes in sphingolipid metabolism is shown in Figure 7. The expression of 3-dehydrosphinganine reductase (KDSR) was up-regulated in the pathway of sphinganine. The gene expression of sphinganine C4-monooxygenase (SUR2) which catalyzed the synthesis of phytosphingosine and phytoceramide, was enhanced. Acid ceramidase (ASAH1), which catalyzes phytoceramide to form phytosphingosine, was up-regulated. Very-long-chain ceramide synthase (LAG1, 2.3.1.297), promoting the mutual conversion between sphinganine and dihydroceramide, was increased. The glucosylceramidase (GBA, 3.2.1.45) gene expression of synthetic *N*-Acylsphingosine was promoted. Alkaline ceramidase (ASAH3, 3.5.1.23) and sphingoid base *N*-stearoyltransferase (CERS1, 2.3.1.299), which promoted the conversion between ceramide and sphingosine, were up-regulated [27].

### 2.9. The Expression of a Key Protein in Autophagy Was Up-Regulated in the KEGG Pathway

Phagocytosis is a basic biological defense mechanism. Phagosomes play an important role in sequestering cytoplasmic components which are delivered to the vacuole for breakdown. The differences of gene expression between transgenic duckweed Pen3a and WT in the phagosome metabolic pathway were studied (Figure 8). Phagolysosomes can degrade antigens to form immunogenic peptides. V-type H+-transporting ATPase subunit A (V-ATPase) gene expression was up-regulated in three stages of phagolysosome formation. Tubulin alpha (TUBA), an important protein in microtubules, was up-regulated after the expression of the penaeidins 3a gene. The expression of protein transport protein Sec61 subunit alpha (Sec61A) was up-regulated. The Sec61 complex promotes the ubiquitination of misfolded peptide chains or unassembled protein subunits. These results effectively confirmed the enhanced antibacterial ability of transgenic duckweed Pen3a.

### 2.10. Proteomics Analysis of WT and Pen3a Duckweed

A label-free algorithm was used to label the proteomic data quantitatively. As in Figure 9a, in the process of protein cluster analysis, Pen3a and WT samples and quantitative information of proteins were classified. The protein clustering results of the Pen3a vs. WT comparison group show a significant difference. In the significant difference analysis of quantitative results, the proteins that meet the screening criteria of expression difference multiplier greater than 1.5 times and *p*-value < 0.05 in the three repeated experimental data in the sample group were selected as significantly differently expressed proteins. We used the protein expression difference multiplier (fold change) between the two groups of samples and the *p*-value obtained by the *t*-test to draw the volcano plot. The abscissa of the volcano map is the fold change, and the ordinate is the significance (*p*-value). The volcano plot of the Pen3a vs. WT comparison group shows the relative quantitative information of significant difference proteins in the two groups of samples, in which the red point is the significantly up-regulated protein, the green point is the significantly down-regulated protein, and the gray point is the non-significant difference protein (Figure 9b). To further reveal the function of the proteins, we analyzed the bacterial secretion system and protein export of the protein interaction network (Figure 9c). The dotted and solid lines indicate a confidence score. The default lowest value is 400. The solid line indicates a higher value and the dotted line indicates a lower value. Round nodes represent proteins/genes and red represents up-regulation and green represents down-regulation. Rectangular nodes represent a KEGG pathway/biological process, the significance *p*-value is represented by the yellow-blue gradient, and yellow color means *p*-value is low, blue means *p*-value is high. Taking the KEGG pathway as the unit and all qualitative proteins as the background, the significance level of protein enrichment of each pathway was analyzed and calculated by Fisher’s exact test, to determine the significantly affected metabolic and signal transduction pathways. As Figure 9d shows, different colors represent different classifications of metabolic pathways. Among them, the difference in protein enrichment in the metabolic pathway was the most significant.

## 3. Discussion

### 3.1. Duckweed in Aquaculture Feed and Wastewater Treatment

The duckweed offers the production of valuable animal feed ingredients [28]. In this study, the duckweed, as well as the Pen3a duckweed, provide a high protein content. The results showed that the protein content of Pen3a duckweed was as high as 37.4% (Table 1). All the amino acids were present in duckweed. Additionally, the adequate quantity of amino acids in duckweed made it a new generation sustainable crop [29]. This was consistent with the results suggesting that duckweed offered the production of valuable animal feed ingredients [28]. In our study, the transcriptome study in Pen3 duckweed vs. WT duckweed showed the most up-regulated DEGs of “protein metabolic process” in the BP section, and the most up-regulated DEGs of “ion binding” in the MF category (Figure 6). The proteomics analysis showed that protein enrichment in the metabolic pathway was the most significant (Figure 9). These results explained the improved protein content in Pen3 duckweed. Furthermore, the enhanced biomass accumulation was consistent with the increased P and K absorption. With a P removal rate of 45.795 ± 0.0164 mg/m^2^/day and a K removal rate of 255.086 ± 0.0264 mg/m^2^/day (Table 1), the Pen3a duckweed could be able to treat aquaculture wastewater. Because the aquaculture wastewater was rich in N, P, and K, and without other pollution, the duckweed could be cultivated in wastewater for producing aquaculture feed.

### 3.2. Duckweed Applied as a Bioreactor

Antimicrobial peptides, including penaeidins, are constituted by divergent classes of peptide isoforms in shrimp, which are the terminal immunity effectors in immunity response [30]. Some strategies have been built to control certain diseases in shrimp aquaculture. For example, penaeidins have been found to play a role in restricting white spot syndrome virus infection by antagonizing the envelope proteins to block viral entry [22]. Two penaeidin isoforms from *Litopenaeus vannamei*, named Lva-PEN 2 and Lva-PEN 3, respectively, strongly bound to bacteria and possessed antiproteinase activity [21]. In this study, gene expression analysis showed that the genes involved with the sphinganine, phagocytosis, and autophagy pathways have been up-regulated significantly. Sphinganine plays an important role in the basic biological defense mechanism. Phagocytosis is the basic biological defense in plants during stress. The differences of gene expression between transgenic duckweed Pen3a and WT suggested the reason for the improved growth situation in Pen3a duckweed treated with wastewater.

Duckweed provides an advantage to express a foreign protein. For example, the M2e peptide of avian influenza virus H5N1 had a high-yield expression in transgenic duckweed, promising the development of a duckweed-based expression system to produce an edible vaccine against avian influenza, such as high-yield expression of m2e peptide of avian influenza virus H5N1 in transgenic duckweed plants. Additionally, duckweed was considered a promising source of protein for food products due to its high protein content and environmentally friendly production (Duckweed as human food. The influence of meal context and information on duckweed acceptability of Dutch consumers). Here, we report a workable way to express penaeidins in duckweed. The Pen3a duckweed could be both cultured in industrial and pond conditions, making use of the shrimp-culturing wastewater. Duckweed, co-cultured with shrimp wastewater in an outdoor water recirculation system, could be applied to obtain penaeidins extraction, feed production, and play a role in bacteria inhibition.

### 3.3. Pen3a Transgenic Duckweed in Controlling the Bacterial

Through the establishment of an efficient duckweed transformation system, transgenic antimicrobial peptide Pen3a plants were successfully obtained, and the antimicrobial peptide duckweed was tested. The bacteriostatic test showed that Pen3a_K duckweed extract inhibited *E. coli* and *S. aureus*, which can be applied to inhibit bacteria. It has been reported that duckweed hosts a similar bacterial assemblage as the terrestrial leaf microbiome in taxonomy [31]. Duckweed is also a kind of traditional Chinese medicine. The quantitative proteomics cluster volcano plot also changed significantly. Additionally, this study demonstrated bacterial secretion system and protein export of the protein interaction network (Figure 9c). The microbiome study in this work showed the roles of duckweed, as well as Pen3a transgenic duckweed, in controlling the bacterial abundance and effectively inhibiting the growth of *Nitrospirae*. The histogram in this study can explain the greatest differences between microbial communities. Scientists have proven that duckweed can be cultured in the wastewater produced from pig raising for nutrient recovery and biomass production [32]. We also investigated the tolerance and the growth of duckweed in the lake water. As shown in Figure 3, the Pen3a duckweed shows a decreased root abscission and an enhanced frond condition, suggesting a better growth in the lake. Hence, the design of Pen3a transgenic duckweed could be cultured in labs, industrial conditions, and native lakes, with important technological applications.

## 4. Materials and Methods

### 4.1. Construction of Antimicrobial Peptide VPS Vector

Penaeidins 3a (Pen3a) cDNA was obtained from RNA of *Litopenaeus vannensis* by reverse transcription as a template. The product was connected with the CaMV-35S promoter and NOS terminator of tobacco mosaic virus by PCR and enzyme digestion. After double enzyme (*Bgl* II and *Pst* I) digestion, the plant expression vector p1301-Pen 3a was constructed.

### 4.2. Duckweed Culture Condition

The experimental material *Lemna turionifera* 5511 was collected from Xiqing District, Tianjin. The culture method refers to the liquid medium described by Wang and Kandeler [33]. Briefly, the liquid culture medium consisted of 0.4 mM MgSO_4_·7H_2_O, 1.4 mM Ca(NO_3_)_2_·4H_2_O, 1.0 mM KNO_3_, 0.4 mM KH_2_PO_4_, 0.4 mM Mg(NO_3_)_2_·6H_2_O, 50 µM CaCl_2_·2H_2_O, 50 µM KCl, 6.1 µM Na_2_MoO_4_·2H_2_O, 69 µM H_2_BO_3_, 30 µM K_2_H_2_EDTA·2H_2_O, 56.7 µM FeNH_4_-EDTA, 13.8 µM MnCl_2_·4H_2_O, 2.8 µM ZnNa_2_EDTA·4H_2_O, 4.8 µM CoSO_4_·7H_2_O, 18.6µM Na_2_-EDTA·2H_2_O, and the pH was adjusted to 5.8. Duckweed was cultured at 23 ± 2 °C with a light intensity of 95 µmol m^−2^ s^−1^ and the photoperiod was 16 h of light and 8 h of darkness.

### 4.3. Plant Transformation and GUS Activity Detection

For the methods of callus induction and regeneration of duckweed, we referred to Yang et al. (2012) [11]. Briefly, the p1301-pen 3a vector was transformed to Agrobacterium tumefaciens EHA105. The duckweed callus was co-cultured with Agrobacterium (OD600 = 0.6) and vacuumed for 30 s 3 times, then cultured with slow shaking at 28 °C for 30 min under dark conditions. The callus were transferred to B5 medium at 28 °C and cultivated under dark conditions overnight. The callus were cultivated on a selection medium containing 1 mg L^−1^ Hygromycin and 160 mg L^−1^ Cefalexin for 3 days then transferred to regeneration medium containing 1 mg L^−1^ Hygromycin, 160 mg L^−1^ Cefalexin, and 0.1 M Serine Cultivate for 30 days until the callus regenerated. Histochemicalstable b-glucuronidase (GUS) stain was performed to determine the transformation of duckweed using Biosharp GUS Staining Kit (Biosharp, Hefei, China; according to instruction).

### 4.4. Establishment of N, P and K Adsorption System for Duckweed

WT and Pen3a duckweed were cultured in fish wastewater for 7 days. Then, the fresh duckweeds were dried at 65 °C until the weight stopped changing, and the dry weight (DW) was measured. Total phosphorus (TP), total nitrogen (TN) and total kalium (TK) in duckweeds were analyzed by the Beijing QingXi Technology Research Institute. In brief, N and P were determined by inductive coupled plasma emission spectrometer (ICP, Agilent 7500a, Santa Clara, CA, USA), as by described Hansen. Additionally, the samples of N levels were obtained by elemental analyzer (Elementar, EA, vario MACRO cube, Frankfurt, Germany).

### 4.5. RNA Sequencing

#### 4.5.1. RNA Isolation and Quantification

RNA samples from duckweed callus were extracted using a Tiangen kit (Tiangen RNA simple Total RNA kit, TIANGEN, Washington, DC, USA). mRNA was purified from total RNA using poly-T oligo-attached magnetic beads and detected on 1% agarose gel to ensure the sample quality. The purity of RNA was measured with the Nanophotometer (IMPLEN, Munich, Germany). RNA concentration was analyzed using Qubit^®^ RNA Assay Kit in Qubit^®^ 2.0 Fluorometer (Life Technologies, Waltham, MA, USA). RNA integrity was assessed using the RNA Nano 6000 assay kit of the Agilent BioAnalyzer 2100 system (Agilent Technologies, Santa Clara, CA, USA). A total amount of 1.5 µg RNA was used for Library preparation for Transcriptome sequencing. Sequencing libraries were generated using NEBNext^®^ Ultra™ RNA Library Prep Kit for Illumina^®®^ (NEB, Ipswich, MA, USA).

#### 4.5.2. Sequencing Data Filtering and Transcript Assembly

Data images of sequencing fragments measured by high-throughput sequencers were transformed into sequence data (readings) by CASAVA base recognition. The raw data obtained from sequencing included a small number of reads with sequencing adaptors or low sequencing quality. Filtered content: removed adapters; removed reads whose proportion of N is greater than 10%; removed low-quality reads. The clean reads were assembled by the Trinity de novo assembly program with min_kmer_cov set to 2 by default, otherwise, it was defaulted [34]. Overall, a reference sequence with an average length of 1928 bp and a total length of 282527137 bp was obtained for subsequent analysis. The combination comparison of differential clustering gene analysis could obtain the differential gene sets, and the FPKM values of the union of all comparative combinations of differential gene sets in the six samples was used for hierarchical clustering analysis.

#### 4.5.3. DNA Extraction and Polymerase Chain Reaction (PCR)

Total DNA was isolated from duckweed using TAKARA MiniBEST Plant Genomic DNA Extraction Kit (TAKARA, Tokyo, Japan; according to the manufacturer’s instruction). DNA concentration was quantified by NanoDrop spectrophotometer. Experimental setup and execution were conducted using a Veriti^®^ 96-Well Thermal cycler, according to the protocol provided by the manufacturer (ABI, Thermo Fisher Scientific, Waltham, MA, USA). PCR products were analyzed by agar gel electrophoresis. The forward primers and reverse primers are TACGCGGAGCACCAGACGGA and TCAACCGGAATATCCCTTT.

### 4.6. Bacteriomic and Bioinformatics Analysis

Firstly, raw data FASTQ files were imported into a format which could be operated by the QIIME2 system using qiime tools import program. Then, the QIIME2 DADA2 plug-in was used for quality control, pruning, denoising, splicing, and removal of chimera to obtain the final feature sequence table. The QIIME2 feature-classifier plugin was then used to align ASV sequences to a pre-trained GREEN GENES 13_8 99% database (trimmed to the V3V4 region bound by the 338F/806R primer pair) to generate the taxonomy table. Any contaminating mitochondrial and chloroplast sequences were filtered using the QIIME2 feature-table plugin. Secondly, ANCOM, ANOVA, Kruskal–Wallis, LEfSe and DEseq2 were used to identify the bacteria that differed in abundance between the groups and samples. Thirdly, diversity metrics were calculated using the core-diversity plugin within QIIME2. Feature level alpha diversity indices, such as observed OTUs, Chao1 richness estimator, Shannon diversity index, and Faith’s phylogenetic diversity index were calculated to estimate the microbial diversity within an individual sample. Beta diversity index was used to assess the differences in microbial community structure among the samples and was subsequently demonstrated by PCoA and NMDS maps [35]. Finally, redundancy analysis (RDA) was performed to reveal the association of microbial communities concerning environmental factors based on the relative abundances of microbial species at different taxa levels using the R package “vegan”.

### 4.7. Antibacterial Test of Transgenic Duckweed Extract

Pen3a duckweed, cultured for 14 days, was frozen in liquid nitrogen and thoroughly ground. Then, it was diluted with 2 g to 5 mL phosphate-buffered saline (PBS, PH 7.2–7.4, 10 mM). The inoculation rings were used to pick the strains on the inclined surface of the test tube in sterile water with glass beads, and the spores were dispersed by shaking with hands for several minutes. The mixed spore suspension was prepared after filtration. The bacteria liquid was mixed with 15–20 mL melted agar medium, then it was self-cooled. The filter paper was soaked in antimicrobial peptides and placed in the center of the plate with bacteria on it. After culturing for 2–3 days, the presence and size of the bacteriostatic circle around the filter paper were observed.

### 4.8. Label-Free Quantification Proteomics

Protein was extracted from tissue samples using SDT lysis buffer (4% SDS, 100 mM DTT, 100 mM Tris-HCl pH 8.0), and was boiled for 5 min before further ultrasonication, and then boiled again for another 5 min. Undissolved cellular debris was removed by centrifugation at 16,000g for 15 min. The supernatant was collected and quantified with a BCA Protein Assay Kit (Bio-Rad, Hercules, CA, USA). Protein digestion (200 µg for each sample) was performed with the FASP method [36]. Then, LC-MS/MS analysis was performed. LC-MS/MS was performed on a Q Exactive Plus mass spectrometer coupled with Easy 1200 nLC (Thermo Fisher Scientific). Sequence database search and data analysis: The MS data were analyzed using MaxQuant software version 1.6.0.16. MS data were searched in the UniProtKB Rattus norvegicus database (36,080 total entries, downloaded 14 August 2018.). The quantitative protein ratios were weighted and normalized by the median ratio in Maxquant software. Only proteins with fold change ≥1.5 folds and a *p*-value < 0.05 were considered as significantly differential expressions. Bioinformatics analysis of bioinformatics data was carried out with Perseus software (v2.0.9.0) [37].

## 5. Conclusions

Our study demonstrated that Pen3a duckweed showed its potential in aquacultural wastewater treatment, feed production, and bacteria inhibition. The transcriptome and proteomics analyses were studied in the WT duckweed and transgenic Pen3a duckweed. The expression of sphingolipid metabolism and phagocytosis process-related genes were up-regulated in Pen3a duckweed. The quantitative proteomics also changed significantly. According to that, Pen3a duckweed had a higher tolerance to the aquacultural wastewater, and a greater colony inhibition ability. In the future, the Pen3a duckweed can be cultured in the lab, under industrial conditions, and in local aquacultural wastewater treatment ponds. Additionally, it has many possible applications in wastewater removal, as raw material for feed production, and for antimicrobial peptides production.

## Figures and Tables

**Figure 1 plants-12-01715-f001:**
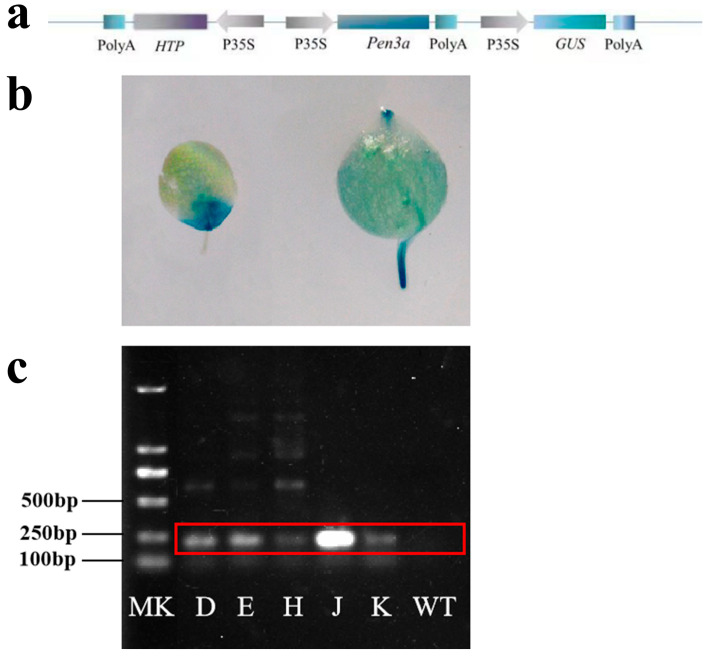
(**a**) T-DNA schematic representation of the final plasmid pCAMBIA-1301-Pen3a. Poly A: the terminator of the HPT gene and the GUS terminator; HTP: hygromycin resistance gene; P35S: the promoter of CaMV-35S; Pen3a: penaeidin 3a gene; GUS: β-glucuronidase gene; Identification of duckweed by GUS staining and RT-PCR. (**b**) GUS activity in transgenic lines of Pen3a duckweed. (**c**) RT-PCR identification of the Pen3a duckweed. MK represented marker; WT was wild-type duckweed; D, E, H, J, K were Pen3a_D, Pen3a_E, Pen3a_H, Pen3a_J, Pen3a_K duckweed.

**Figure 2 plants-12-01715-f002:**
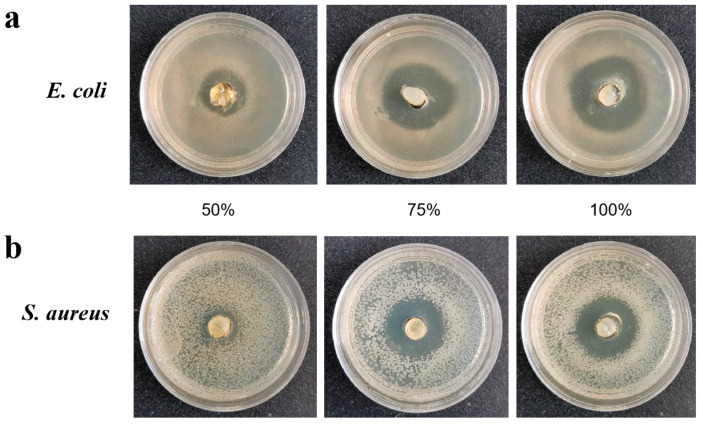
Zone of inhibition of various bacteria by Pen3a duckweed extracting solution. (**a**) Inhibition zone of different concentrations Pen3a duckweed extracting solution on *Escherichia coli*. (**b**) Inhibition zone of different concentrations Pen3a duckweed extracting solution on *Staphylococcus aureus*.

**Figure 3 plants-12-01715-f003:**
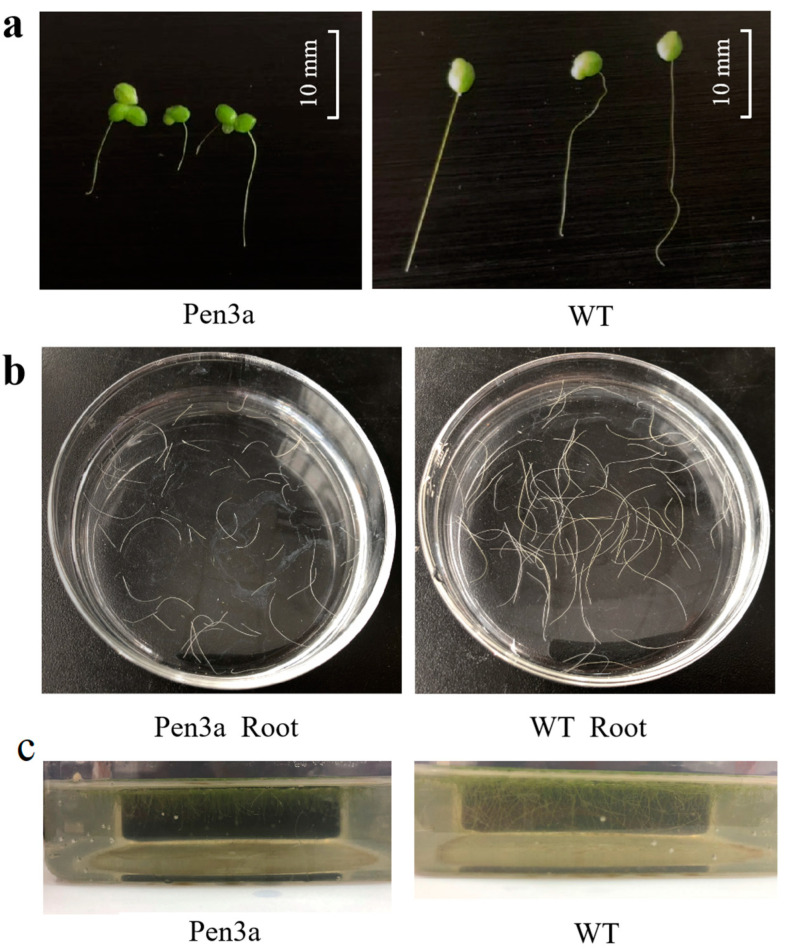
The phenotype of duckweed cultured in aquaculture lake water for 14 days. (**a**) Difference of root length between WT and Pen3a duckweed after 14 days of culture. Scale bar = 10 mm; (**b**) root abscission rate of WT and Pen3a duckweed; (**c**) Pen3a and WT water body conditions after 14 days of culture in the same cultured aquaculture wastewater.

**Figure 4 plants-12-01715-f004:**
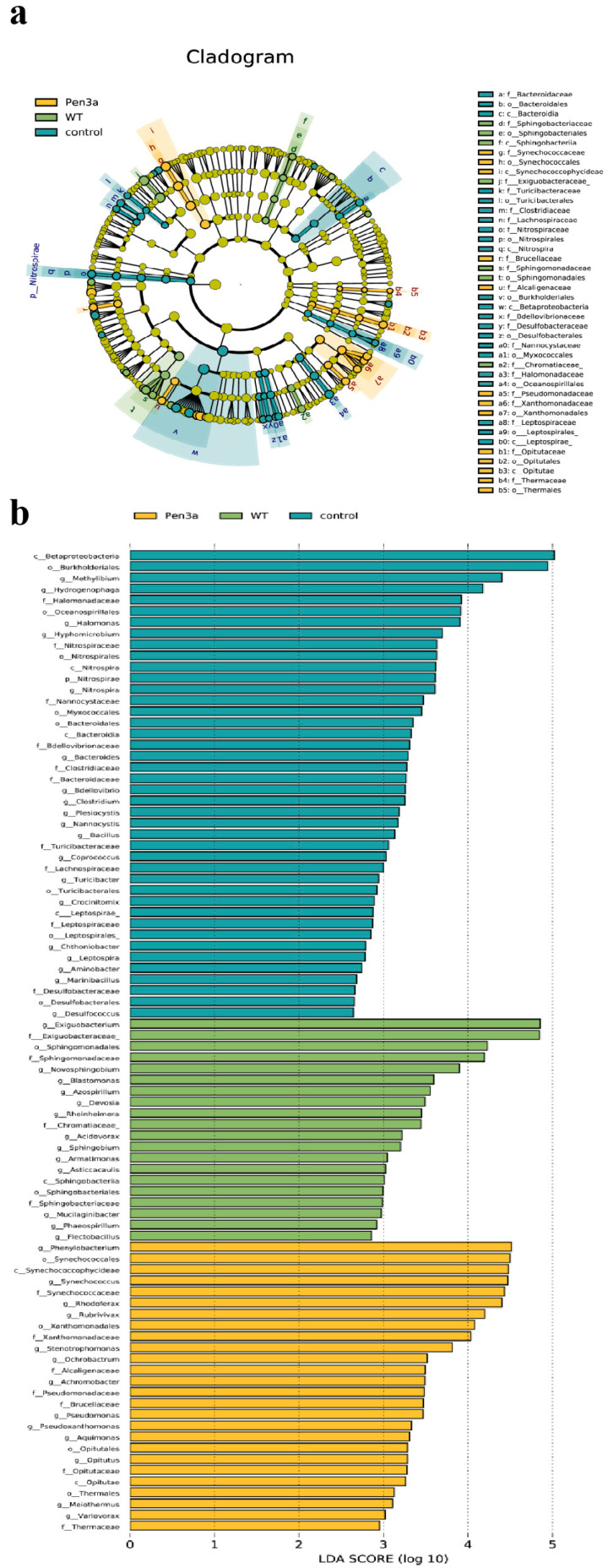
(**a**) LEfSe results on aquart microbial communities. Cladogram corresponds to different taxonomic levels of families and genera in the phylum. Each circle node represents a species. The yellow node represents no significant difference between groups, and the non-yellow is the characteristic microorganism of the corresponding color group and the abundance is significantly higher in this group. Blue indicates the control group, green indicates wild-type treatment, yellow indicates transgenic antimicrobial peptide duckweed treatment. (**b**) Histogram of the LDA scores computed for abundance between three treatments. Each horizontal bar represents a species, and the length of the bar corresponds to the LDA scores. The higher the LDA value, the greater the difference. The LDA scores of indicator microbiota of the three types of sediments were greater than 2.5 logs 10.

**Figure 5 plants-12-01715-f005:**
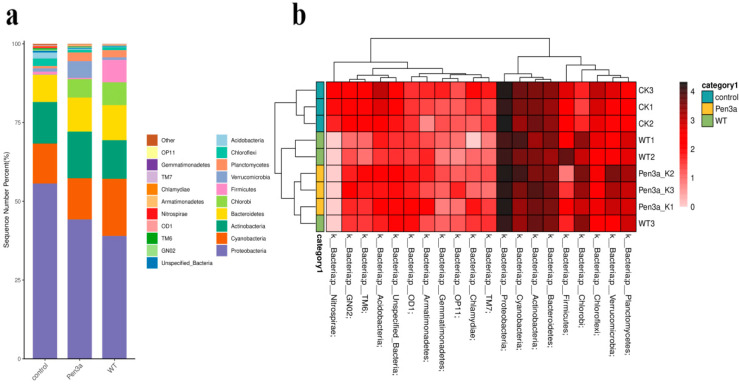
Abundance of the bacterial community in aquaculture water after 14 days treatment. (**a**) Phylum taxonomic distribution. The abscissa represents three processing methods, and the ordinate represents the ratio of the number of series at the level of the gate to the total annotation data. It means the top 20 microbial phyla with relative abundance. (**b**) Heatmap of microbial community structures at the phylum level.

**Figure 6 plants-12-01715-f006:**
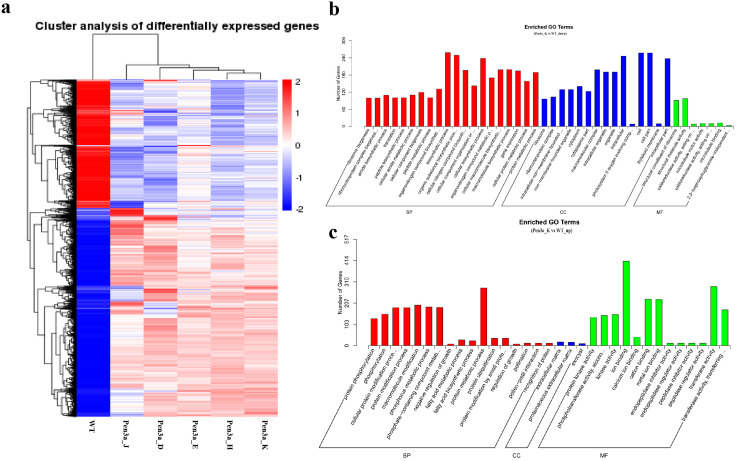
(**a**) Heat map of cluster analysis of DGEs in six genotype duckweed. (**b**) Go enrichment histogram of down-regulated differential genes. Enriched GO Terms of Pen3a_K versus WT. (**c**) Go enrichment histogram of up-regulated differential genes.

**Figure 7 plants-12-01715-f007:**
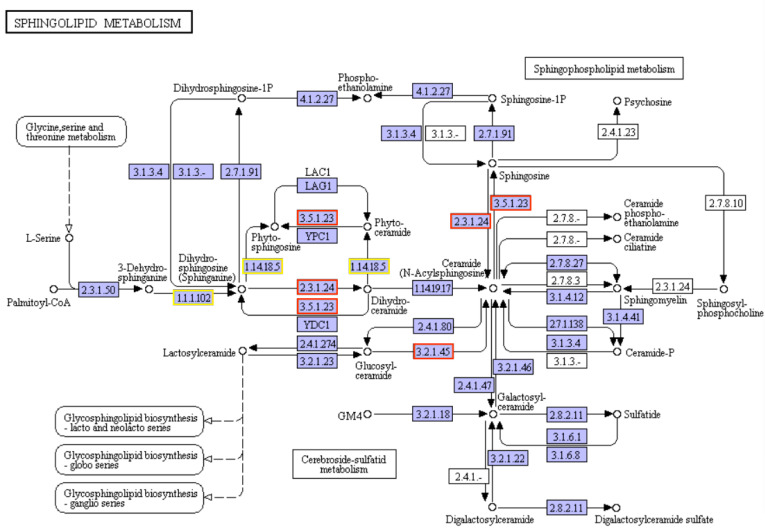
Gene changes in sphingolipid metabolism KEGG pathway. Changes in gene expression of sphingolipid metabolism pathway in KEGG pathway of Pen3a duckweed compared with WT duckweed. Red box shows the enhanced expression, and yellow box shows the mixed regulation of expression. Green box shows the decreased expression (no enzyme has been down-regulated in this figure).

**Figure 8 plants-12-01715-f008:**
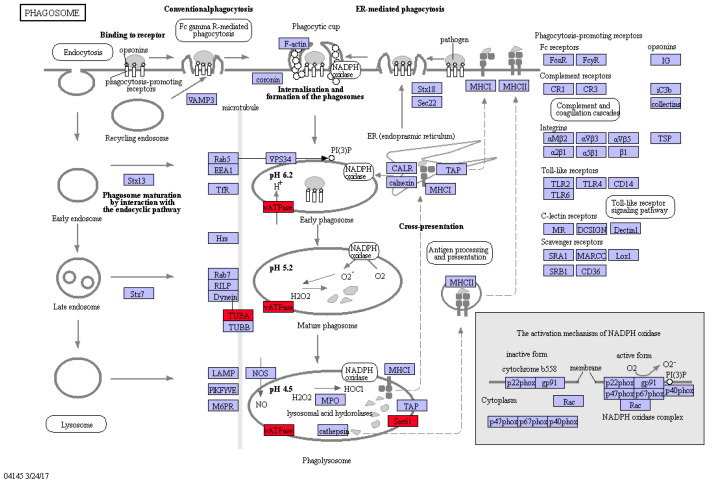
Changes of duckweed phagosome KEGG pathway between Pen3a and WT. The block’s color meant unigenes coding corresponding proteins, determined by the expression pattern. The red blocks indicated the up-regulated genes.

**Figure 9 plants-12-01715-f009:**
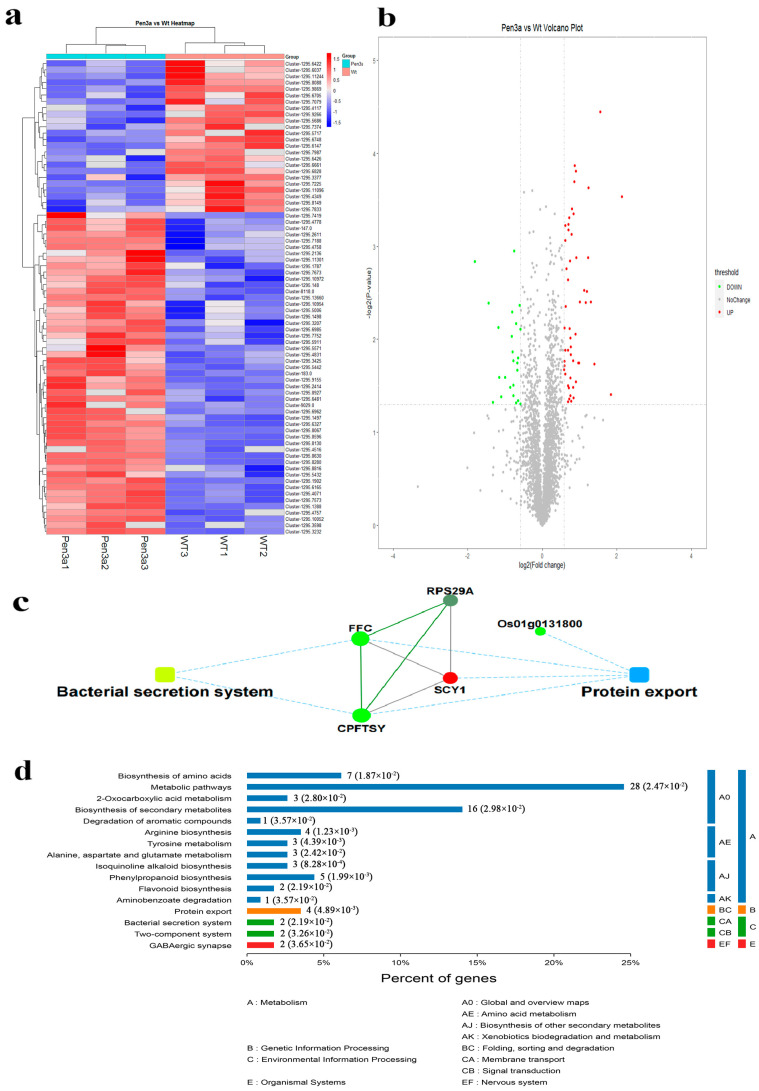
Proteomic analysis. (**a**) Results of hierarchical cluster analysis of expressed proteins to Pen3a and WT comparison group. The protein abundance was shown as color code, blue means lower abundance and red means higher abundance. Abundance difference factor in protein hierarchy clustering *p* < 0.05. (**b**) Volcano Plot between Pen3a and WT comparable group. The red dots were significantly up-regulated proteins, the green dots were significantly down-regulated proteins, and the gray dots were non-significantly different proteins. (**c**) Interaction network results of differentially expressed proteins in Pen3a vs. WT comparison group. (**d**) KEGG enrichment pathway statistics of Pen3a vs. WT comparison group.

**Table 1 plants-12-01715-t001:** Production of duckweeds (dry wight).

Content	Unit	WT	Pen3a
Protein (Pro)	%	36.27 ± 0.079	37.38 ± 0.102
Ash	%	9.17 ± 0.045	8.57 ± 0.041
Nitrogen (N)	%	5.80 ± 0.012	5.98 ± 0.016
Phosphorus (P)	%	1.39 ± 0.0005	1.10 ± 0.0002
Potassium (K)	%	7.74 ± 0.0008	6.86 ± 0.0003
N removal rate	mg/m^2^/day	191.15 ± 0.395	197.082 ± 0.527
P removal rate	mg/m^2^/day	45.795 ± 0.0164	36.24 ± 0.0067
K removal rate	mg/m^2^/day	255.086 ± 0.0264	220.152 ± 0.0096

## Data Availability

The original contributions presented in the study are included in the article; further inquiries can be directed to the corresponding author.

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
