# Peer review of "The Antimicrobial Potential and Aquaculture Wastewater Treatment Ability of Penaeidins 3a Transgenic Duckweed"

_plants, 2023, doi:10.3390/plants12081715_

Round 1

Reviewer 1 Report

The paper presents an interesting study that employs duckweed as a feedstock for aquatic wastewater treatment and the production of antimicrobial peptides. However, some improvements could be made to enhance the clarity and persuasiveness of the study. Specifically:

- All scientific names should be italicized throughout the manuscript.

- For the antibacterial test, it would be more persuasive to include the WT duckweed extracting solution as a control. Additionally, the reason for selecting E. coli and S. aureus for this experiment should be explained. It would also be interesting to test the effects of Pen3a duckweed on some of Nitrospirae, as mentioned in the text.

- In Section 2.4, the authors mention that the water treated by Pen3a duckweed was "very clean," but this should be shown in quantitative terms for clarity and persuasiveness.

- Figure 7 needs clarification on the difference between the yellow boxes and red boxes. The figure legend should be modified to explain this difference.

Reviewer 2 Report

The authors generated transgenic duckweed (Lemna turionifera 5511) expressing Penaeidins 3a, an antimicrobial peptide. The transgenic plant is likely to be tolerant against waste water. They checked the microbe communitiy, transcriptome and proteome and discovered  differences between the transgenic plant and wild-type. However, the differences were not discussed in the Discuccsion section. Thus, the molecular analyses in the later part are too descriptive and seem not to contribute to understand the effect of  Penaeidins 3a. This manuscript should be improved to integrate  those interesting results

In result section, starting with a brief introduction of the experimental design may be improove readability. For example, readers cannot imagine Penaeidins 3a expressing plants have GUS activity.

There are many minor errors throughout the manuscript, such as, no italic scientific name, two P removal rate in Table1. 

Round 2

Reviewer 1 Report

All of the points I raised have been thoroughly and effectively addressed. The author has done an good job of incorporating my feedback and ensuring that the paper is of the highest quality possible.